# Effect of H-ZSM-5 and Al-MCM-41 Proportions in Catalyst Mixtures on the Composition of Bio-Oil in Ex-Situ Catalytic Pyrolysis of Lignocellulose Biomass

**Devy Kartika Ratnasari [1],\*** , **Anton Bijl [2]**, **Weihong Yang [1]** and **Pär Göran Jönsson [1]**

[1]   KTH-Royal Institute of Technology, Brinellvägen 23, 10044 Stockholm, Sweden; weihong@kth.se (W.Y.); parj@kth.se (P.G.J.)

[2]   Alucha Management B. V., Industriepark Kleefse Waard, 6827 AV Arnhem, The Netherlands; anton.bijl@alucha.com

\*   Correspondence: dkra@kth.se

**Abstract:** The present work is an attempt to optimize the proportion of H-ZSM-5 and Al-MCM-41 in the catalyst mixtures for lignocellulose biomass catalytic pyrolysis. The H-ZSM-5 proportions of 50.0, 66.7, 75.0, and 87.5 wt.% were examined for the upgrading of biomass pyrolysis vapors in the fixed bed reactor. The catalyst mixture of 87.5 wt.% H-ZSM-5 and 12.5 wt.% Al-MCM-41 was found most effective in this study, giving a 65.75% deoxygenation degree. An organic-rich bio-oil was obtained with 74.90 wt.% of carbon content, 8 wt.% of hydrogen content, 15 wt.% oxygen content, a 0.39 wt.% water content, and a high heating value of 34.15 MJ/kg. The highest amount of desirable compounds among the studied catalytic experiments, which include hydrocarbons, phenols, furans, and alcohols, was obtained with a value of 95.89%. A significant improvement in the quality of bio-oil with the utilization of H-ZSM-5 and Al-MCM-41 catalyst mixtures was the rise of desirable compounds in bio-oil.

**Keywords:** Al-MCM-41; catalyst; H-ZSM-5; lignocellulose; pyrolysis

---

## 1. Introduction

As a source of carbon for liquid transportation fuels, biomass must be converted to liquids at high yields of high quality [1,2]. One of the promising biomass materials for transportation fuel replacement is lignocellulose biomass. The availability of lignocellulose biomass is abundant [3]. It is also known to be the cheapest and fastest-growing of the biomass materials [4]. Further, it is a scalable, economically viable, and potential carbon neutral feedstock for the production of bio-oil [5].

A challenge for the efficient conversion of biomass is to produce fuels which are compatible with the existing refinery and internal combustion engines [1]. In response to the challenge of producing biofuels, pyrolysis has received consideration as a method for transforming biomass into liquid fuel.

Pyrolysis refers to a thermal degradation process of biomass with a high temperature in the absence of oxygen. The process is followed by rapid cooling to form a liquid pyrolysis oil (hereinafter 'bio-oil') [1]. Bio-oil contains highly oxygenated compounds and a high concentration of water, which causes bio-oil to retain particular properties. Besides having high viscosity and acidity, bio-oil has poor storage stability, is thermally unstable, has a low heating value, and suffers from poor mixing with fossil fuels [1,2,6,7]. The elimination of oxygen is thus necessary to transform bio-oil into a liquid fuel that would be broadly accepted and economically attractive [6]. The use of heterogeneous catalysis in

biomass pyrolysis is regarded as a possible technology to boost bio-oil quality and expand the use of bio-oil.

A concept from the petrochemical industry is taken for the biomass pyrolysis process, where the heterogeneous catalyst is used to convert the heavy fractions into light fuels and chemicals [8]. The catalytic pyrolysis of biomass is a thermochemical technology to upgrade the quality of bio-oil [8] either by removing oxygenates [6,7] and undesirable compounds, such as acids and carbonyls [2], or by increasing the production of specific compounds, such as benzene, toluene, and xylene (BTX). Lappas et al. claimed that an ideal catalyst should promote the biomass pyrolysis process to produce not only a high yield of bio-oil, but also a high-quality product with low oxygen and water contents [8]. Ultimately, the target of the process is to produce a bio-oil that could be used either directly as a liquid fuel or as a drop-in feedstock (or co-feedstock) in modern refineries, much like fossil crude oil.

H-ZSM-5 catalyst has been widely investigated as a catalyst for oil and fuel production as well as for the production of renewable oil and chemicals from biomass [9,10]. Due to the size of the micropores [11], the shape selectivity of pore structure [12], and its strong acidity [11,13], H-ZSM-5 catalyst induces changes in the composition of bio-oil by reducing the amount of oxygenated compound via deoxygenation reaction [11,13,14] and promoting the formation of aromatic hydrocarbons [11–13,15]. However, H-ZSM-5 catalyst is also known for its high coking formation and rapid catalyst deactivation [11,12,15,16]. Some high-molecular-weight oxygenates, which cannot enter the pores of microporous H-ZSM-5 catalyst, polymerize and form coke on the catalyst surface [9,12,17,18].

On the other hand, the amorphous aluminosilicates with ordered mesoporous structures, a large pore size, and mild acidity can convert the high-molecular-weight oxygenates to a less bulky compound [9], give a high organic yield, as well as reduce the chances of coke deposition and pore-blocking [13,17,19]. Al-MCM-41 is one of the mesoporous materials with high surface area and more accessible reaction sites than the traditional H-ZSM-5 catalyst [17,19]. However, products may escape before complete pyrolysis because the pore size is too large [18]. In addition, catalytic pyrolysis of biomass using Al-MCM-41 catalyst results in the formation of polycyclic aromatic hydrocarbons (PAHs), which is not a desirable product in the bio-oil [17,19,20].

Considering the advantages of H-ZSM-5 and Al-MCM-41, researchers have altered the shape selectivity and porosity of catalysts to allow multi-step/cascade reactions to take place [21–24] in order to render a proper design of catalytic pyrolysis process of biomass and to provide high activity, selectivity, and longer life catalyst. Araujo et al. [25] investigated the catalytic effect of mechanical mixtures of H-ZSM-5 and Al-MCM-41 on the quality of bio-oil from sunflower oil. They found that the mixture of 50% H-ZSM-5 and 50% Al-MCM-41 resulted in a balance carbon fraction between gasoline ($C_5$–$C_{10}$), kerosene ($C_{11}$–$C_{15}$), and diesel ($C_{16}$–$C_{24}$) ranges, corresponding to 44.7%, 42.2%, and 10.5%, respectively [25]. More recently, Li et al. [9] studied the online upgrading of vacuum pyrolysis of rape straw in a two-stage reactor using H-ZSM-5 and MCM-41. They found that 1:1 mixed ratio of H-ZSM-5:MCM-41 was the best among all ratios in their experiments. The oxygen content, H/C, O/C, the high heating value of the bio-oil were 12.81%, 1.701, 0.126, and 34.31 MJ/kg, respectively [9]. Other studies have been carried out on the mixture of microporous and mesoporous catalysts apart from H-ZSM-5 and Al-MCM-41 [25–27]. Nevertheless, the synthesized catalysts have been of limited use on the bench-scale reactor. In industrial production, factors which influence the preference for catalysts in the industry are the ease of separation of products from catalysts [28–30], reusability [29], relatively inexpensive [29,30], and ease to handle [29]. In this study, a simple method is proposed to improve the quality of bio-oil during the biomass catalytic pyrolysis process by physically mixing commercial mesoporous catalyst in pellet form, Al-MCM-41, and microporous catalyst in sphere form, H-ZSM-5. This approach can provide an affordable process that can be easily adapted in industries.

A kinetic study of the H-ZSM-5/Al-MCM-41 catalyst mixture and its effect on lignocellulose biomass pyrolysis was conducted in previous research [31]. The study found that the catalyst mixtures had a notable impact on the fractions of volatile substances from lignocellulose biomass. Also, the energy activation for lignocellulose biomass was decreased when using the catalyst mixtures was used [31].

Moreover, the authors performed experimental investigations on the effect of catalyst mixtures on the lignocellulose biomass pyrolysis using a bench-scale fixed bed reactor [32]. The Al-MCM-41 with H-ZSM-5 in the staged catalyst method has been shown to facilitate the synthesis of desirable compounds, such as hydrocarbons, phenols, furans, and alcohols. Ex-situ catalytic pyrolysis of beech wood at a temperature of 500 °C with H-ZSM-5:Al-MCM-41 ratio of 3:1 resulted in a bio-oil containing 11.08 wt.% of organic liquid products and 76.20 wt.% of desirable fractions [32].

The present study is a continuation of the previous works in an attempt to verify an optimum proportion of H-ZSM-5 and Al-MCM-41 in the catalyst mixture to produce a high quality of bio-oil with a low content of oxygen and a high amount of desirable compounds during the ex-situ catalytic pyrolysis of lignocellulose biomass. The goal of this study is to provide more detailed investigations with emphases on the elemental compositions and the heating values of the bio-oil. The degree of deoxygenation and the catalyst characterizations before and after the catalytic pyrolysis process were not addressed in prior works.

## 2. Results and Discussion

A physical mixture of mesoporous catalyst, Al-MCM-41, and microporous catalyst, H-ZSM-5 was used to produce bio-oil with a low content of oxygen and a high amount of desirable compounds from the catalytic pyrolysis of lignocellulose biomass. In the previous work, Ratnasari et al. [32] studied the effect of two-stage catalyst bed on the bio-oil quality. Although more product is formed when using two-stage catalyst bed, due to increased reactant/catalyst contact [32–35], the catalyst is difficult to replace [34] as the entire reactor has to be dismantled for the replacement of catalyst [35] and the unit may be difficult to service and clean [36]. The purpose of using physically mixed catalysts in this study was to provide an affordable process that can be easily adapted in industries, which mostly use fluidized bed reactor [37,38] for continuous process and catalyst regeneration [39].

### 2.1. Effect of H-ZSM-5 and Al-MCM-41 Proportions on Product Distribution

The product distribution for all catalytic pyrolysis with different proportions of H-ZSM-5 and Al-MCM-41 is presented in Table 1. The error bars represent standard deviations in absolute % for all experiments. The standard deviations were calculated by Excel with a non-biased or n-1 method [40]. The scatter in the product yields is always less than 5%, indicating that the reproducibility is sufficient for observing trends in all experiments. The organic fraction (OF) yield increased with an increased H-ZSM-5 proportion in the catalyst. Among the catalytic pyrolysis tests with different proportions of H-ZSM-5 and Al-MCM-41, the highest OF value was achieved with H87.5/A12.5 catalyst. The OF yield obtained with H87.5/A12.5 was 5.66 wt.%, and the corresponding aqueous fraction (AF) yield was 36.40 wt.%. It is noteworthy that the H-ZSM-5 catalyst favours the transformation of pyrolysis vapours to OF. Smaller oil yields are associated with more deoxygenation—the higher the deoxygenation, the lower the mass yield [41]. Chen et al. [42] also found that H-ZSM-5 produced higher oil yields than when using an MCM-41 catalyst. Those results are consistent with the results of the current study.

The highest yield of non-condensable gases among the catalytic pyrolysis with catalyst mixtures was reached when using an H87.5/A12.5 catalyst, and the value amounted to 13.36 wt.%. The trend for gaseous products over the catalyst mixture is in the lines of the results of earlier literature data [42], which showed that H-ZSM-5 produced more gases than when using Al-MCM-41. Consequently, as the proportion of H-ZSM-5 in the catalyst mixture increased, the non-condensable gases rose. The active sites of H-ZSM-5 favour the secondary cracking reactions of hydrocarbon and generate organic liquid fractions and enhance the formation of gas [43]. The detailed composition of non-condensable gases can be found in the Supplementary Table S2.

**Table 1.** Yields of Organic Fraction (OF) in liquid, Aqueous Fraction (AF) in liquid, non-condensable gases, and coke on catalysts in wt.% for the non-catalytic and catalytic pyrolysis with different proportions of H-ZSM-5 and Al-MCM-41.

| Experiment | OF | AF | Gas | Coke | Char | Total |
|---|---|---|---|---|---|---|
| NC * | 24.60 ± 0.04 | 23.35 ± 0.03 | 10.09 ± 0.01 | 0.00 | 40.40 ± 0.02 | 98.44 ± 0.04 |
| H-ZSM-5 | 7.08 ± 0.04 | 33.48 ± 0.03 | 13.92 ± 0.05 | 2.78 ± 0.01 | 40.25 ± 0.02 | 97.50 ± 0.02 |
| H87.5/A12.5 | 5.66 ± 0.03 | 36.40 ± 0.03 | 13.36 ± 0.05 | 2.22 ± 0.02 | 40.25 ± 0.05 | 97.89 ± 0.04 |
| H75/A25 | 4.25 ± 0.05 | 37.63 ± 0.05 | 11.80 ± 0.05 | 2.38 ± 0.04 | 40.48 ± 0.01 | 96.54 ± 0.03 |
| H66.7/A33.3 | 3.54 ± 0.01 | 40.24 ± 0.04 | 11.64 ± 0.02 | 2.89 ± 0.01 | 40.14 ± 0.02 | 98.44 ± 0.04 |
| H50/A50 | 2.83 ± 0.02 | 42.28 ± 0.05 | 11.41 ± 0.02 | 3.22 ± 0.04 | 40.17 ± 0.03 | 99.92 ± 0.04 |
| Al-MCM-41 | 1.42 ± 0.05 | 44.08 ± 0.02 | 10.19 ± 0.01 | 3.72 ± 0.02 | 40.15 ± 0.04 | 99.56 ± 0.01 |

* NC: non-catalytic case, OF+AF in one phase: AF is water fraction.

The coke yield decreased from 3.72 wt.% when using single Al-MCM-41 catalyst to 2.22 wt.% when 87.5 wt.% of H-ZSM-5 catalyst was added in the catalyst mixtures. Then, the coke yield increased to 2.78 wt.% when using single H-ZSM-5 catalyst. The fluctuation of coke yield when using single H-ZSM-5 catalyst was caused by the microporosity structure of H-ZSM-5, which hindered the large molecule oxygenates to enter the pores and caused the growth of coke [9,12,17,18]. In the absence of Al-MCM-41 catalyst, there was no pathway to convert large oxygenates to small ones before reacting with H-ZSM-5 catalyst. However, the yield of coke in Al-MCM-41 catalyst was 0.94 wt.% higher than it in H-ZSM-5 catalyst. This result is contradicted by the experiments of Zhang et al. [26] on catalytic pyrolysis of rice stalk using LOSA-1, which mainly composed of ZSM-5, and Gamma-$Al_2O_3$. The coke yield in their experiment decreased from 30.3% with pure LOSA-1 to 23.5% with 50% Gamma-$Al_2O_3$. Gamma-$Al_2O_3$, as a mesoporous catalyst, had strong cracking characteristic and can convert the large-molecule oxygenates into small-molecule oxygenates. Thus, the oxygenates molecules can leave the catalyst after their formation before they polymerize and lead to coke [44].

Nonetheless, the current finding is consistent with the results of past studies by Adjaye et al. [45]. They observed that the coke resulting from a biofuel conversion, when using silica–alumina (6.8–27.9 wt.%), a mesoporous catalyst, was higher compared to when using an H-ZSM-5 catalyst (2.2–14.1 wt.%) [45]. Twaiq et al. [46] also found that the amount of coke formed from catalytic pyrolysis of palm oil was higher over MCM-41 compared to ZSM-5 and USY zeolites. With MCM-41, the coke yield was 5–12 wt.%, whereas with HZSM-5, it was about 2, and 5 wt.% with USY zeolite under the same reaction conditions [46]. In this study, the high coke deposition in Al-MCM-41 catalyst might be due to their low acidity and high pore volume, resulting in the inefficient dehydrogenation of pyrolysis vapor and the formation of coke [47]. Aguado et al. [48] added that due to Al-MCM-41 uniform mesoporosity, the formation of coke is feasible in Al-MCM-41 unhindered porous structure [48]. Kim et al. [49] also showed a similar result that the catalytic activity of Al-MCM-41 decreased faster than H-ZSM-5 due to its low acidity and coking. Among the experiments with catalyst mixture, the coke yield was reduced with an increased H-ZSM-5 proportion. The lowest yield among all the catalytic pyrolysis was 2.22 wt.% when using H87.5/A12.5, which suggested that there was a synergistic effect between H-ZSM-5 and Al-MCM-41 catalysts.

The catalyst did not have any influence on the char yields since the char was recovered before the catalyst bed, and there was no physical mixing of the biomass and catalyst. Hence, the catalyst did not influence the primary conversion of the biomass. Consistent with the findings by Bakar and Titiloye [42], the char after the reactions were around 40 wt.% in this study.

### 2.1.1. Elemental Composition of Liquid Products

The elemental composition and HHV of the OF and AF are presented in Table 2. The elemental analysis of the non-catalytic oil reflects the typically high content of oxygen found for biomass-derived pyrolysis oils, in comparison to the analysis of OF in catalytic oils.

**Table 2.** Elemental composition of Organic Fraction (OF) and Aqueous Fraction (AF) in liquid (wt.%), the deoxygenation degree (%), and the High Heating Value (HHV) in OF for non-catalytic and catalytic pyrolysis with different proportions of H-ZSM-5 and Al-MCM-41.

| Experiment | Organic Fraction (OF) | | | | | | Aqueous Fraction (AF) | | | | |
|---|---|---|---|---|---|---|---|---|---|---|---|
| | Elemental Composition (wt.%) | | | | Deoxygenation Degree (%) | HHV (MJ/kg) | Elemental Composition (wt.%) | | | | HHV (MJ/kg) |
| | C | H | O | N | | | C | H | O | N | |
| Non-catalytic | 47.25 | 7.91 | 43.10 | 0.50 | 1.60 | 19.55 | | | | | |
| H-ZSM-5 | 73.50 | 7.79 | 21.00 | 0.61 | 52.50 | 32.28 | 5.91 | 10.70 | 60.00 | 0.94 | 6.53 |
| H87.5/A12.5 | 74.90 | 8.00 | 15.00 | 0.59 | 65.75 | 34.15 | 5.59 | 11.00 | 78.10 | 0.50 | 3.57 |
| H75/A25 | 82.05 | 8.37 | 8.60 | 0.50 | 80.37 | 38.26 | 3.74 | 11.20 | 67.70 | 0.50 | 1.28 |
| H66.7/A33.3 | 80.00 | 8.11 | 10.30 | 0.61 | 76.48 | 36.89 | 3.37 | 11.10 | 54.80 | 0.50 | 7.20 |
| H50/A50 | 84.20 | 8.24 | 6.70 | 0.62 | 84.70 | 39.15 | 2.58 | 11.10 | 73.50 | 0.50 | 3.53 |
| Al-MCM-41 | 85.60 | 8.20 | 4.80 | 0.63 | 89.04 | 39.91 | 2.37 | 11.10 | 79.40 | 0.50 | 2.39 |

In the analysis of OF, Table 2 shows that the carbon content of the bio-oil increased as the proportion of Al-MCM-41 in the catalyst mixtures increased. The highest hydrogen content of all the catalytic pyrolysis experiments was observed when using H75/A25, and it corresponded to 8.37 wt.%. Contrarily, with an increased proportion of Al-MCM-41 in the catalyst mixture, the oxygen content decreased.

As illustrated in Figure 1a, the position of the oils obtained from catalytic pyrolysis is lower (lower H/C ratio) and in the left (lower O/C) when compared to the non-catalytic experiment. During the upgrading of bio-oil vapours, oxygen may be released through a dehydration reaction, resulting in water production [50,51]. Decarbonization and decarboxylation prompt the formation of CO and $CO_2$, respectively [51]. Low H/C and O/C ratio reveal that the oils produced from catalytic pyrolysis experiments retain higher quality than the non-catalytic oil, according to Choi et al. [52]. The calculated High Heating Values (HHVs) of catalytic oils reaffirmed the high quality of catalytic oils, which were 12.73–20.36 MJ/kg higher than that of the non-catalytic pyrolysis oil (19.55 MJ/kg). The oil produced from catalytic pyrolysis experiments with a catalyst mixture H87.5/A12.5 had the highest H/C ratio, which is a favourable property of fuel [52,53]. Later, it can also be seen that the catalytic H87.5/A12.5 oil had more desirable compounds compared to other experiment cases.

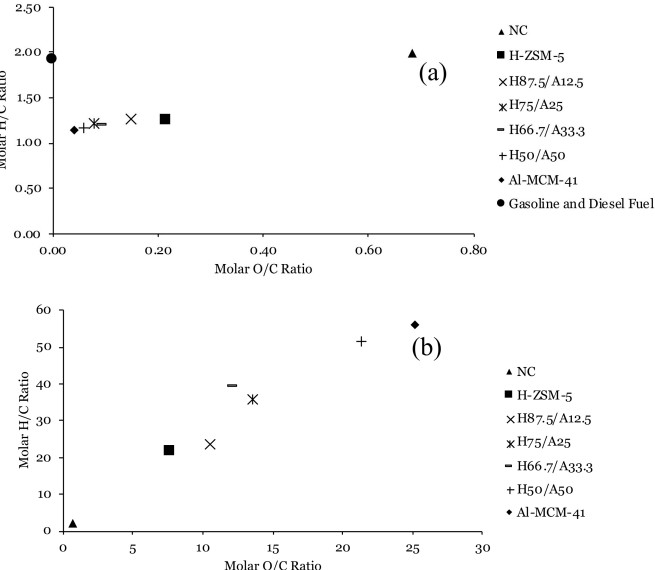

**Figure 1.** The van Krevelen plot of (**a**) Organic Fraction (OF); (**b**) Aqueous Fraction (AF) produced from pyrolysis of lignocellulose biomass without catalyst (non-catalytic) and with catalyst mixtures. A gasoline and diesel fuel case (54) (in (**a**)) is included for comparison.

On the other hand, the trend of elemental composition for AF was in contrast to the results for OF. It appears from Table 2 that the carbon content in the AF decreased as the proportion of Al-MCM-41 in

the catalyst mixtures increased. Further, the addition of Al-MCM-41 in the catalyst mixture did not alter the content of hydrogen in the AF. There was a fluctuation in the oxygen content of AF.

Figure 1b illustrates that the position of the AFs obtained from all cases of catalytic pyrolysis is higher (higher H/C ratio) and located to the right (higher O/C) when compared to the non-catalytic experiment. High H/C and O/C ratio implied that the AFs had no better quality than the non-catalytic oil. The calculated HHV values of AFs are also lower than that of the non-catalytic pyrolysis oil. Even though the AFs had high values of the H/C ratio, the high O/C ratio decreased the higher heating values (HHVs) of the AFs as compared to a non-catalytic oil [55].

The deoxygenation degree in OFs and AFs was also evaluated. As shown in Table 2, a high level of deoxygenation (84.70 wt.%) was achieved as the oxygen content of the bio-oil in the catalytic pyrolysis with catalyst mixtures decreased to 6.70 wt.% when using an H50/A50 catalyst. With an increased Al-MCM-41 proportion in the catalyst mixture, the deoxygenation degree increased. A small proportion of Al-MCM-41 in the H87.5/A12.5 promoted the removal of oxygen, resulting in the deoxygenation degree 13.25% higher than for a single H-ZSM-5 catalyst. The change might be a result of the absence of acid sites on the Al-MCM-41 catalyst as well as the mesoporosity of it, which inhibits oxygen removal of pyrolysis vapours [56–58]. Furthermore, the proportion of H-ZSM-5 catalysts promotes greater extents of deoxygenation reactions [19,59]. For the AFs, the calculated values of the deoxygenation degree were negative, as the AFs contained oxygen levels 1.27–1.84 times higher compared to those for a non-catalytic oil.

### 2.1.2. High Heating Value of Liquid Products

The High Heating Values (HHVs) in OF are shown in Table 2. As a comparison, the heating values of conventional fuel oil are in the range of 40–43 MJ/kg, according to Nguyen et al. [60] and Huber et al. [61]. An increase of the proportion of H-ZSM-5 in the catalyst mixture resulted in energy losses to gas. The catalyst mixtures application had a significant effect on the deoxygenation degree and HHV of bio-oil.

The HHVs of AFs in MJ/kg are presented in Table 2. Overall, there is no clear trend observed for the HHV of AFs. From Figure 1b, it is seen that the oxygen levels in the AFs increase as the proportion of H-ZSM-5 in the catalyst mixture decrease. It is generally considered that the increase of the oxygen content leads to a decrease of the HHV [62–65]. The oxygen content in fuel will reduce the ability of the fuel to burn, resulting in less amount of heat produced and low heating value. The fluctuating high heating values (HHVs) are to occur due to the high water content in AFs, which interferes in the elemental analysis.

### 2.1.3. Water Content in Liquid Products

Figure 2 shows the influence of the catalyst mixtures on the water content in the OF of liquid products. The AF from the catalytic pyrolysis of biomass consisting of water amounted to 86.10–95.01 wt.%. Similarly, earlier studies [66–68] also reported high water contents in the AF. With a decreased proportion of H-ZSM-5 in the catalyst mixture, the water content increased. Water originates from a dehydration reaction of organic compounds in addition to the physically bonded water present in the biomass [66].

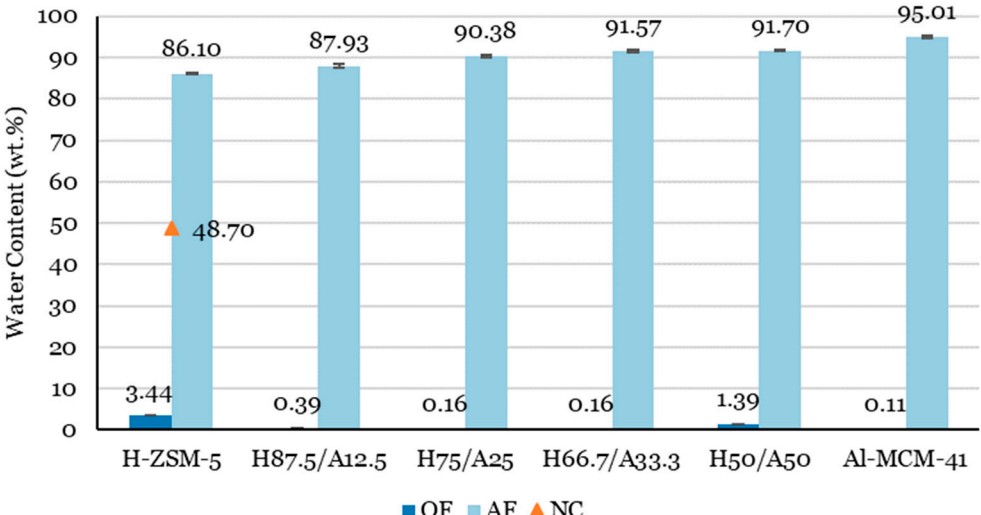

**Figure 2.** Influence of H-ZSM-5 and Al-MCM-41 proportions on the water content in the OF and AF for lignocellulose biomass catalytic pyrolysis. A result from a non-catalytic case (orange triangle) is included for comparison.

### 2.1.4. Chemical Composition of Liquid Products

The chemical compositions of liquids from non-catalytic and catalytic pyrolysis of lignocellulose biomass are summarised in Table 3. In the analysis of OF, the lowest selectivity of benzene (10.91%), toluene (7.20%), and xylene (7.48%) were obtained when using a single Al-MCM-41 catalyst. The BTX yields were correlated with strong acid contents of the catalysts [32,69], resulting in the high catalytic activity in aromatization [70]. The data in Table 3 indicates that H-ZSM-5 had higher strong acid content than Al-MCM-41. Therefore, among the experiments with catalyst mixture, the selectivity of BTX decreased with a further decreased proportion of H-ZSM-5 in the catalyst mixture.

The present findings also highlight the selectivity of toluene and xylene, which was higher when using H87.5/A12.5 than when using a single H-ZSM-5 catalyst. Moreover, the highest selectivity of toluene (12.64%) and xylene (26.42%) was obtained when using H87.5/A12.5 catalyst mixture. A synergistic effect was observed for the formation of toluene and xylene, which could be explained by the enhanced Diels-Alder type reactions between furans and olefins from biomass [26,49]. This could be attributed to the cracking effect of Al-MCM-41 on the biomass for H-ZSM-5 conversion. The catalyst mixture of H-ZSM-5 and Al-MCM-41 can alter the selectivity of aromatics in the OF.

Table 3 shows that furans, naphthalenes, and PAHs increased as the proportion of H-ZSM-5 in the catalyst mixture decreased. The rise in PAHs is also influenced by the lignin content, as lignin tends to produce more polyaromatics [71]. A dehydration reaction of levoglucosan from cellulose and anhydrosugar from hemicellulose might explain the increased furan yield. A reaction network for lignocellulosic biomass catalytic pyrolysis is proposed and shown in Figure 3.

**Table 3.** Detailed chemical compositions (area%) of the Organic Fraction (OF) and Aqueous Fraction (AF) for catalytic pyrolysis of lignocellulose biomass.

| Sample | Catalytic Liquids | | | | | | | | | | | |
|---|---|---|---|---|---|---|---|---|---|---|---|---|
| | Organic Fraction | | | | | | Aqueous Fraction | | | | | |
| | H-ZSM-5 | H87.5/A12.5 | H75/A25 | H66.7/A33.3 | H50/A50 | Al-MCM-41 | H-ZSM-5 | H87.5/A12.5 | H75/A25 | H66.7/A33.3 | H50/A50 | Al-MCM-41 |
| *Aliphatics selectivity (area%)* | | | | | | | | | | | | |
| Paraffins | 0.00 | 0.00 | 0.00 | 0.00 | 0.00 | 0.00 | 0.25 | 0.00 | 0.00 | 0.00 | 0.00 | 0.00 |
| Iso-Alkanes | 0.00 | 0.00 | 0.00 | 0.00 | 0.00 | 0.00 | 0.00 | 0.00 | 0.00 | 0.00 | 0.00 | 0.00 |
| Cyclo-Alkanes (Naphtene series) | 0.51 | 0.00 | 0.44 | 0.44 | 0.00 | 0.29 | 0.00 | 0.00 | 0.00 | 0.00 | 0.00 | 0.00 |
| Olefins | 0.00 | 0.00 | 0.18 | 0.10 | 0.00 | 0.00 | 0.00 | 0.00 | 0.00 | 0.00 | 0.00 | 0.00 |
| *Total Aliphatics* | *0.51* | *0.00* | *0.62* | *0.54* | *0.00* | *0.29* | *0.25* | *0.00* | *0.00* | *0.00* | *0.00* | *0.00* |
| *Aromatics selectivity (area%)* | | | | | | | | | | | | |
| Benzene and derivatives | 14.73 | 13.23 | 13.74 | 12.95 | 11.40 | 10.91 | 9.32 | 2.58 | 0.90 | 1.18 | 0.00 | 0.00 |
| Toluene | 11.37 | 12.64 | 11.14 | 9.48 | 8.26 | 7.20 | 1.26 | 0.00 | 0.00 | 0.00 | 0.00 | 0.00 |
| Xylene | 20.50 | 26.42 | 24.38 | 18.86 | 11.35 | 7.48 | 0.17 | 0.00 | 0.00 | 0.00 | 0.00 | 0.00 |
| Naphthalene | 11.58 | 11.86 | 13.88 | 18.01 | 18.30 | 19.21 | 0.00 | 0.00 | 0.00 | 0.00 | 0.00 | 0.00 |
| Styrene | 0.27 | 0.27 | 0.25 | 0.21 | 0.20 | 0.18 | 0.00 | 0.00 | 0.00 | 0.00 | 0.00 | 0.00 |
| Polycyclic Aromatic Hs (PAHs) | 2.02 | 1.69 | 2.56 | 5.56 | 6.55 | 7.08 | 0.32 | 0.43 | 0.40 | 0.31 | 0.39 | 0.00 |
| Other aromatics | 11.50 | 10.06 | 8.88 | 9.25 | 7.25 | 6.64 | 5.46 | 6.00 | 3.83 | 3.41 | 2.40 | 0.00 |
| *Total Aromatics* | *71.96* | *76.17* | *74.82* | *74.32* | *63.31* | *58.70* | *16.52* | *9.01* | *5.13* | *4.90* | *2.79* | *0.00* |
| *Oxygenates (area%)* | | | | | | | | | | | | |
| Alcohols | 0.00 | 0.00 | 0.00 | 0.00 | 1.06 | 0.48 | 0.18 | 0.21 | 0.00 | 0.00 | 0.00 | 0.00 |
| Ethers | 0.00 | 0.00 | 0.00 | 0.00 | 0.82 | 1.23 | 0.00 | 0.00 | 0.00 | 0.00 | 0.00 | 0.00 |
| Aldehydes | 0.00 | 0.00 | 0.00 | 0.00 | 1.54 | 2.21 | 0.14 | 1.69 | 1.11 | 0.00 | 0.00 | 0.00 |
| Ketones | 1.08 | 1.55 | 0.86 | 0.49 | 0.39 | 0.54 | 8.17 | 8.73 | 7.20 | 5.22 | 6.00 | 6.68 |
| Acids and Esters | 0.00 | 0.00 | 0.00 | 0.00 | 1.63 | 1.53 | 16.89 | 18.54 | 20.03 | 21.92 | 18.42 | 13.02 |
| Phenols | 21.89 | 20.32 | 19.91 | 20.07 | 22.91 | 23.73 | 45.22 | 53.48 | 62.93 | 64.88 | 71.81 | 78.70 |
| Furans | 1.56 | 1.10 | 1.59 | 2.20 | 4.28 | 4.88 | 0.00 | 0.24 | 0.00 | 1.52 | 0.00 | 0.00 |
| Carboxylic | 0.22 | 0.10 | 0.00 | 0.00 | 0.13 | 1.37 | 0.36 | 0.16 | 0.67 | 0.00 | 0.00 | 0.00 |
| Amines and Amides | 2.78 | 0.56 | 1.41 | 1.50 | 2.18 | 2.88 | 1.48 | 1.35 | 1.81 | 1.57 | 0.98 | 1.60 |
| *Total Oxygenates* | *27.52* | *23.62* | *23.78* | *24.25* | *34.94* | *38.85* | *72.44* | *84.39* | *93.75* | *95.10* | *97.21* | *100.00* |
| *Others (area%)* | 0.00 | 0.21 | 0.78 | 0.89 | 1.75 | 2.16 | 10.79 | 6.60 | 1.12 | 0.00 | 0.00 | 0.00 |

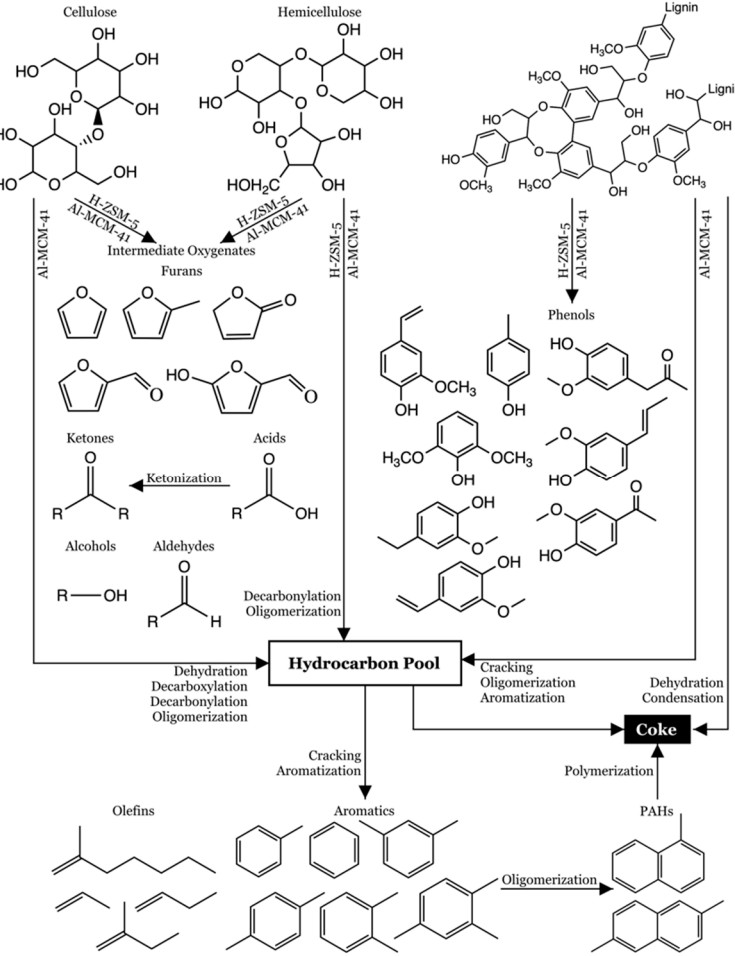

**Figure 3.** Reaction pathways for lignocellulose biomass catalytic pyrolysis with an H-ZSM-5 and Al-MCM-41 catalyst mixture (adapted from Chi et al. [72], Jeenpadiphat et al. [73], and Wang et al. [71]).

The chemical compounds were grouped as desirable and undesirable compounds to compare how the chemical compounds were distributed between OF and AF. Hydrocarbons, phenols, furans, and alcohols were included as desirable compounds [13,74–78], while other compounds, like acids, carbonyls, carboxylic, ethers, polycyclic aromatic hydrocarbons (PAHs), oxygenates, and other unidentified compounds were known to be undesirable compounds [13,74–76,78]. Figure 4 shows the proportion of desirable and undesirable compounds in OF for non-catalytic and catalytic pyrolysis as the percentage of total ion chromatogram area.

As seen in Figure 4, there is a clear trend in the chemical composition of OF with different catalyst proportions. The most significant observed result was the higher percentage of desirable compounds in the catalytic pyrolysis with catalyst mixtures for H87.5/A12.5 and H75/A25, compared to that in the catalytic pyrolysis when using a single catalyst. The catalyst mixture of H87.5/A12.5 and H75/A25 increase the catalytic activity of H-ZSM-5 and Al-MCM-41. The highest amount of desirable compounds was achieved when using an H87.5/A12.5 catalyst, and it amounted to 95.89%. Accordingly, the undesirable compounds increased as the proportion of Al-MCM-41 in the catalyst mixture increased. The lowest amount (4.11%) of undesirable compounds was also achieved when using an H87.5/A12.5 catalyst. Al-MCM-41 is only able to initiate the first cracking, due to its absence of any shape-selective properties, while the shape-selectivity of H-ZSM-5 catalyst promotes the conversion of intermediate products in addition to primary cracking. Intermediate olefins and paraffin are obtained from the cracking using Al-MCM-41 catalyst.

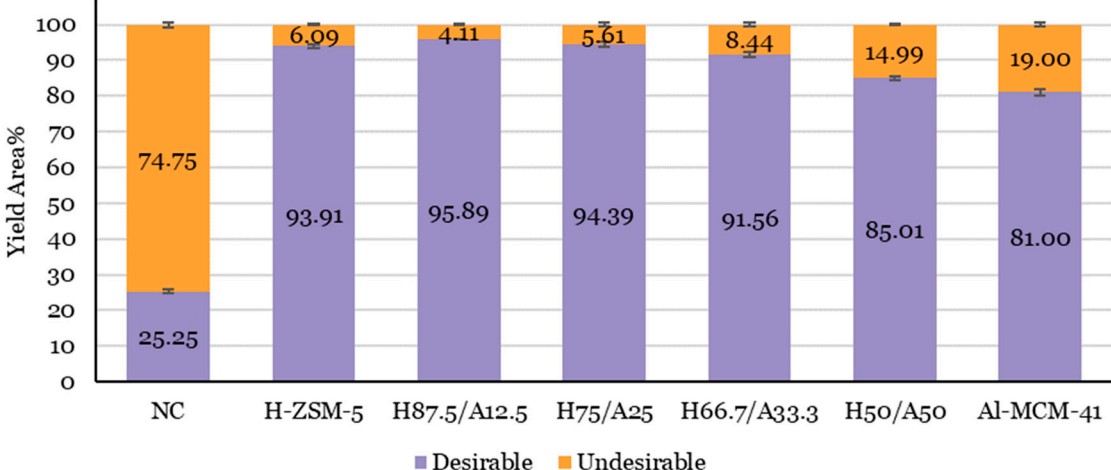

**Figure 4.** The chemical composition (area%, on dry basis (odb)) of OF for lignocellulose biomass catalytic pyrolysis with different proportions of H-ZSM-5 and Al-MCM-41. A non-catalytic case is included as a reference case. Yield Area (%) Desirable = area hydrocarbons (aliphatic, BTX, naphthalene, styrene, other aromatics) % + area phenols % + area furans % + area alcohols %. Yield Area (%) Undesirable = area PAHs % + area ethers % + area aldehydes % + area ketones % + area acids and esters % + area carboxylic % + area amines and amides % + area others (unidentified compounds) %. Data was taken from Table 3.

On the other hand, the strong acid sites and a definite pore structure of H-ZSM-5 promote the formation of aromatic hydrocarbons [45,67] and reduces the number of carbonyls [79] and ketones [67], which determine the acidity of the oil [60,79]. Therefore, the desirable compounds decreased as the proportion of H-ZSM-5 in the catalyst mixture decreased. This result is along the lines of an earlier review published by Kabir and Hameed [19], which showed that the large molecules could access the active catalyst site in the mesoporous catalyst. This easy access to the active catalyst sites can increase the catalytic potential of the microporous zeolite [19].

As shown in Figure 5, a clear trend was observed for the chemical composition of AF from the catalytic pyrolysis of biomass. The highest amount of desirable compounds was 78.70 wt.% when using a single Al-MCM-41 catalyst, followed by 74.21 wt.% when using an H50/A50 catalyst. The number of undesirable compounds decreased with a reduced H-ZSM-5 in the catalyst mixture.

Kechagiopoulos et al. [80] found that the aqueous fraction of bio-oil consists of a complex mixture of oxygenated compounds, including acids, alcohols, aldehydes, and ketones, which is also shown in the results of the present study. Specifically, the number of undesirable compounds in AF increased significantly compared to the amount present in the OF, which support the findings of Li et al. [81] in the catalytic pyrolysis of sawdust.

## 2.2. Catalyst Characterization

The analysis of the BET surface area has been carried out for the fresh and spent catalysts used in the experiments and is illustrated in Figure 6. The BET surface areas were linearly enhanced when Al-MCM-41 catalyst was introduced to the catalyst mixture. The fresh H87.5/A12.5 retained 231.26 m$^2$/g of BET surface area, which was 56.47 m$^2$/g higher compared to a fresh H-ZSM-5 catalyst (174.79 m$^2$/g). The BET surface area increased as the proportion of Al-MCM-41 in the catalyst mixture increased. The fresh Al-MCM-41 possessed the highest BET surface area (537.60 m$^2$/g) among all the catalyst used in the experiment, which implied the mesoporous structure of the catalyst.

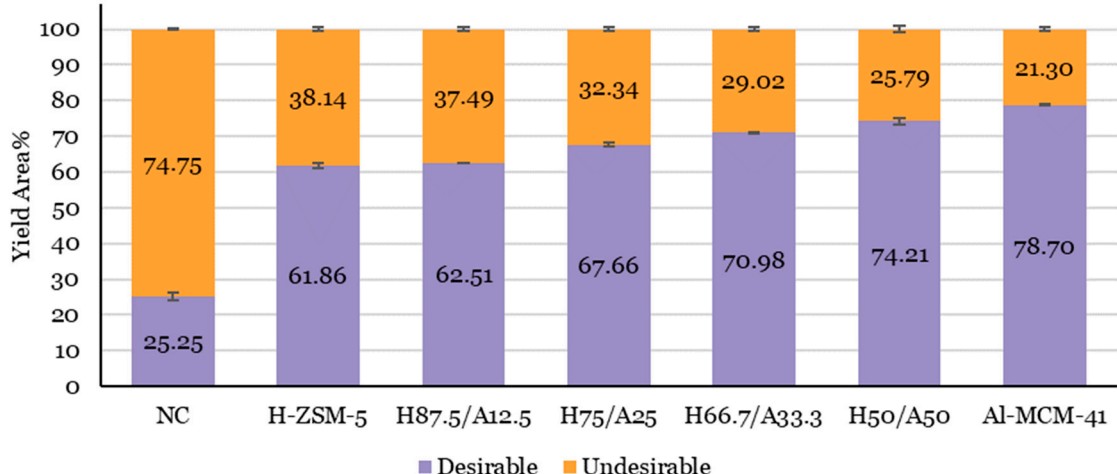

**Figure 5.** The chemical composition of AF for lignocellulose biomass catalytic pyrolysis with different proportions of H-ZSM-5 and Al-MCM-41. A non-catalytic case is included as a reference case. Yield Area (%) Desirable = area hydrocarbons (aliphatic, BTX, naphthalene, styrene, other aromatics) % + area phenols % + area furans % + area alcohols %. Yield Area (%) Undesirable = area PAHs % + area ethers % + area aldehydes % + area ketones % + area acids and esters % + area carboxylic % + area amines and amides % + area others (unidentified compounds) %. Data was taken from Table 3.

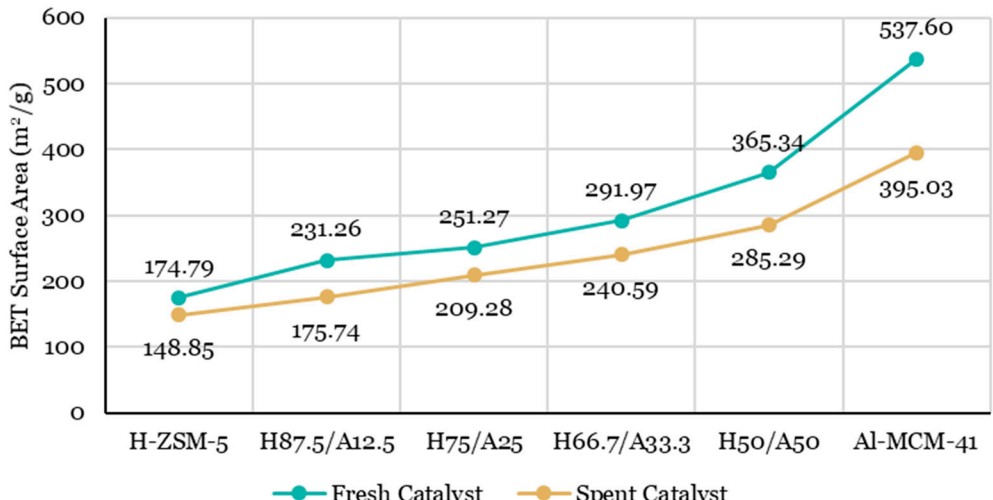

**Figure 6.** The BET surface area of fresh and spent catalysts from catalytic pyrolysis of lignocellulose biomass.

Figure 6 shows that all the BET surface areas of spent catalysts were 14.84–26.52% lower compared to those of fresh catalysts, due to accumulative coke deposition on the catalyst. The largest BET surface area reduction (26.52%), from 537.60 to 395.03 m2/g, was obtained when using a single Al-MCM-41 catalyst.

## 3. Materials and Method

### 3.1. Materials

#### 3.1.1. Biomass

The biomass utilized for the experiments was the industrial biomass of lignocellulose, Lignocel HBS 150–500 by J. Rettenmaier & Söhne GmbH + Co KG, Germany, made from beech wood, with

particle sizes varying from 150 to 500 μm. Before the experiment, the biomass was dried in the oven at 110 °C overnight to reduce the water content. The proximate and ultimate chemical composition of the biomass can be found in the Supplementary Table S1. The analysis was carried out by BELAB AB, Sweden [82].

### 3.1.2. Catalysts

The catalytic materials used for the experiments were the spherical catalyst of H-ZSM-5 (a diameter of 3 mm) and pellet catalyst of Al-MCM-41 (3 mm in diameter by 10 mm in length) both with a Silica-to-Alumina Ratio (Si/Al, SAR) of 25, which were obtained from the Pingxiang Naike Chemical Industry Co. Ltd. and Lingsu Group Co. Limited, China, respectively. The chemical composition in terms of Si/Al ratio was determined by inductively coupled plasma-optical emission spectroscopy (ICP-OES) by the catalyst providers. XRD patterns of the catalysts were given in the Supplementary Figures S1 and S2.

The catalysts were calcined at 550 °C for 15 h, then dried in an oven at 110 °C for 12 h, and stored in a desiccator before the pyrolysis experiments. Catalyst mixtures with H-ZSM-5 proportions of 50.0, 66.7, 75.0, and 87.5 wt.% were used, and they were labelled as H50/A50, H66.7/A33.3, H75/A25, and H87.5/A12.5, respectively. Experiments with single H-ZSM-5 catalyst and single Al-MCM-41 catalyst were also carried out to compare with the experiments using catalyst mixtures.

### *3.2. Methods*

### 3.2.1. Pyrolysis–Catalysis Reactor

The batch experiments were carried out at a temperature of 500 °C using a bench-scale fixed bed reactor. The biomass was introduced into the reactor with a piston system. The catalyst mixture bed (100.00 g) was placed inside the reactor, and biomass (100.00 g) was placed in the piston. A 1:1 biomass:catalyst mixture was kept for all experiments. As soon as the reactor reached 500 °C, the biomass entered the reactor and started to be pyrolyzed. During the pyrolysis process, a constant flow of $N_2$ (350 standard mL/min) was fed from the top of the reactor to maintain an inert atmosphere. The reactor pressure was around ambient. The first 15 min of $N_2$ flow carried the pyrolysis vapors from biomass to pass through the catalyst bed. The $N_2$ flow was kept for another 15 min to enable an additional purging and the withdrawal of the products. A reference experiment without bed material inside the reactor was also performed and labelled NC for non-catalytic in this report. The schematic diagram of catalytic pyrolysis reactor can be found in the Supplementary Figure S3. The pyrolysis–catalysis experiment has been described in more detail by the authors in previous work [32].

All the pyrolysis experiments and analysis were triplicated and performed in random orders to obtain a good estimation of experimental errors, and all the values presented are average values.

### 3.2.2. Product Analysis

Condensable liquid products were collected in five condensers, which were preserved at −15 °C by being kept in an isopropanol bath. The two phases within the liquid products, the organic and aqueous phases, were separated using a decanting funnel. Each phase was analyzed using the elemental analyzer to determine the elemental composition. A Mettler-Toledo volumetric Karl-Fischer titrator T5 was used to determine the water content of each phase.

Gas chromatography Agilent 7890A coupled to a mass spectrometer Agilent 5975C was used to identify the compounds present within the liquid. The column used in the GC method was an HP-5 capillary column of 60 m × 0.250 mm × 0.25 μm for organic fraction analysis and a DB-1701 capillary column of the same size for aqueous fraction analysis. The carrier gas was helium at a constant flow rate of 1 mL/min with splitless injection. The GC/MS injector and interface temperature were maintained at 280 °C. The GC oven temperature was initially set at 45 °C for 2 min. Then, the temperature was



programmed to rise from 45 °C to 230 °C with the heating rate of 20 °C/min and held at 230 °C for 10 min. For the MS measurements, electron ionization (EI) was used with ionization energy of 70 eV. The mass spectra were recorded at 1.49 scans per second with the m/z range of 45–550 amu and ion source temperature of 230 °C. The peak area was obtained by automatic integration with Agilent ChemStation and interpreted using the NIST11 Library with a high degree of certainty (over 80%). The relative proportion of each identified compound was based on its selective ion peak area relative to the total selective ion peak area, following Equation (1), where *i* is the identified compound.

$$\text{Area } i\ (\%) = (\text{area } i/\text{total area})\cdot 100 \tag{1}$$

Non-condensable gases were collected in the Tedlar™ gas bag, which was connected to the condenser system. Then, the gaseous products were then analyzed using a micro-gas chromatograph (GC) Agilent 490. The gas mass was calculated based on the gas chromatography results.

The coke yield on the catalyst surface has been calculated by direct weighting. There was no major shift in the composition of the chars because there was no actual mixing of biomass, and catalysts took place.

The mass yield of catalytic pyrolysis products was calculated according to Equation (2), where *i* is OF, AF, gas, char, and coke.

$$\text{Mass yield } i\ (\text{wt.}\%) = (\text{mass } i\ (\text{g})/\text{biomass (g)})\cdot 100 \tag{2}$$

The higher heating value (HHV) of bio-oil samples was estimated using the Dulong equation [83], where C, H, and O are carbon, hydrogen, and oxygen in weight percentages, respectively.

$$HHV\left(\frac{\text{MJ}}{\text{kg}}\right) = \frac{\left[338.2(\%C) + 1442.8\left((\%H) - \frac{(\%O)}{8}\right)\right]}{1000} \tag{3}$$

The degree of deoxygenation of bio-oils was calculated using Equation (4) [84].

$$\text{Deoxygenation degree } (\%) = \left(1 - \frac{(\%O)_{bio-oil}}{(\%O)_{biomass}}\right)\cdot 100 \tag{4}$$

### 3.2.3. Catalysts Characterization: Surface Area and Porosity Measurements

Catalysts were outgassed at 250 °C for 3 h under vacuum before the tests were carried out. The ASAP 2020 micromeritics instrument was used to calculate the $N_2$ adsorption–desorption and the Brunauer–Emmett–Teller (BET) method was used to assess the surface area.

## 4. Conclusions

Catalytic pyrolysis of lignocellulose biomass using a catalyst mixture with 87.5 wt.% of H-ZSM-5 and 12.5 wt.% of resulted in the highest organic fraction of liquid (5.66 wt.%), the highest yield of non-condensable gases (13.36 wt.%), and the lowest yield of coke (2.22 wt.%). A high heating value (HHV) of 34.15 MJ/kg was achieved with a carbon content of 74.90%, a hydrogen content of 8.00%, and an oxygen content of 15.00%. The desirable compounds amounted to 95.89%, which was also higher compared to catalytic pyrolysis using a single catalyst.

**Supplementary Materials:** The following are available online at http://www.mdpi.com/2073-4344/10/8/868/s1, Table S1: Properties of the Lignocellulosic Biomass used for Catalytic Pyrolysis Experiments on Dry Basis, Figure S1: XRD pattern of H-ZSM-5 (Si/Al = 25) using monochromatic Kα radiation (30–40 kV and 40–50 mA), Figure S2: XRD pattern of Al-MCM-41 (Si/Al = 25), performed at 40 kV and 200 mA with a scan rate of 1 °/min over the range of 1–8°, Figure S3: Schematic Diagram of Catalytic Pyrolysis Reactor, Table S2: Gas composition for pyrolysis and catalytic pyrolysis of lignocellulose biomass (wt.%).

**Author Contributions:** D.K.R.: Conceptualization, Methodology, Software, Validation, Formal Analysis, Investigation, Data Curation, Writing—Original Draft, Visualization, A.B.: Supervision, Conceptualization, Writing—Review and Editing, W.Y.: Supervision, Resources, Writing—Review and Editing, Project Administration, Funding Acquisition, P.G.J.: Supervision, Writing—Review and Editing. All authors have read and agreed to the published version of the manuscript.

**Funding:** This research was funded by the Swedish Energy Agency—Energimydigheten under the ERA-NET Bioenergy program, grant number 43911-1.

**Conflicts of Interest:** The authors declare no conflict of interest.

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
