# Peer review of "Effect of H-ZSM-5 and Al-MCM-41 Proportions in Catalyst Mixtures on the Composition of Bio-Oil in Ex-Situ Catalytic Pyrolysis of Lignocellulose Biomass"

_catalysts, doi:10.3390/catal10080868_

Round 1

Reviewer 1 Report

In this work, Ratnasari et al. studied the effect of H-ZSM-5 and Al-MCM-41 zeolite mixtures on the composition of pyrolysis oil derived from lignocellulose biomass. They have found that the catalyst mixture of 87.5 wt% H-ZSM-5 and 12.5 wt% Al-MCM-41 exhibits the most effective catalytic ability. The results provide valuable information to the CFP field. This work is thus of interest to the audience of the Catalysts. There are several concerns, though, that need to be addressed before acceptance.

  1. In Table 1, OF, AF, and Gas yields all follow a linear tread, i.e. increasing H-ZSM-5 proportion from 0% to 100%, OF and Gas yields increase, whereas AF yield decreases, except for the coke yield. The coke yield decreases gradually from 3.72% to 2.22%, then increases from 2.22% to 2.78% with increasing H-ZSM-5 proportion. Therefore, their statement “On the other hand, the coke yield was decreased with an increased H-ZSM-5 proportion” seems problematic. The authors will need to provide a relevant explanation why the coke yield does not follow a similar linear tread.
  2. In line 200-205, the authors state that coke yield using mesoporous catalyst was higher than that using H-ZSM-5 catalyst, since the H-ZSM-5 constrains coke formation inside the zeolite pores. However, in line 54-57, the authors cited previous literatures demonstrating that mesoporous structures with a large pore size reduce the chances of coke deposition, which contradicts the line 200-205. This needs to be clarified.
  3. Line 79-80, “Nevertheless, few recent studies on the mixture of H-ZSM-5 and Al-MCM catalysts and its application on lignocellulose biomass” is not a completed sentence and missing citations.
  4. Line 81-86 regarding the composition of Lignocel HBS sample use in this work seems irrelevant in this paragraph. The authors may consider moving these sentences to the materials/biomass section (Line 109-113).
  5. Line 46-62, the terms of ZSM-5 and MCM-41, instead of H-ZSM-5 and Al-MCM-41, have been used in these lines.

Author Response

In this work, Ratnasari et al. studied the effect of H-ZSM-5 and Al-MCM-41 zeolite mixtures on the composition of pyrolysis oil derived from lignocellulose biomass. They have found that the catalyst mixture of 87.5 wt% H-ZSM-5 and 12.5 wt% Al-MCM-41 exhibits the most effective catalytic ability. The results provide valuable information to the CFP field. This work is thus of interest to the audience of the Catalysts. There are several concerns, though, that need to be addressed before acceptance.

  1. In Table 1, OF, AF, and Gas yields all follow a linear tread, i.e. increasing H-ZSM-5 proportion from 0% to 100%, OF and Gas yields increase, whereas AF yield decreases, except for the coke yield. The coke yield decreases gradually from 3.72% to 2.22%, then increases from 2.22% to 2.78% with increasing H-ZSM-5 proportion. Therefore, their statement “On the other hand, the coke yield was decreased with an increased H-ZSM-5 proportion” seems problematic. The authors will need to provide a relevant explanation why the coke yield does not follow a similar linear tread.

Response:

As suggested, the paragraphs have been rephrased (line 204-231).

The coke yield decreased from 3.72 wt.% when using single Al-MCM-41 catalyst to 2.22 wt.% when 87.5 wt.% of H-ZSM-5 catalyst was added in the catalyst mixtures. Then, the coke yield increased to 2.78 wt.% when using single H-ZSM-5 catalyst. The fluctuation of coke yield when using single H-ZSM-5 catalyst was caused by the microporosity structure of H-ZSM-5, which hindered the large molecule oxygenates to enter the pores and caused the growth of coke [9,12,17,18]. In the absence of Al-MCM-41 catalyst, there was no pathway to convert large oxygenates to small ones before reacting with H-ZSM-5 catalyst. However, the yield of coke in Al-MCM-41 catalyst was 0.94 wt.% higher than it in H-ZSM-5 catalyst. This result is contradicted by the experiments of Zhang et al. [26] on catalytic pyrolysis of rice stalk using LOSA-1, which mainly composed of ZSM-5, and Gamma-Al2O3. The coke yield in their experiment decreased from 30.3% with pure LOSA-1 to 23.5% with 50% Gamma-Al2O3. Gamma-Al2O3, as a mesoporous catalyst, had strong cracking characteristic and can convert the large-molecule oxygenates into small-molecule oxygenates. Thus, the oxygenates molecules can leave the catalyst after their formation before they polymerize and lead to coke [39].    

Nonetheless, the current finding is consistent with the results of past studies by Adjaye et al. [40]. They observed that the coke resulting from a biofuel conversion, when using a silica-alumina (6.8 – 27.9 wt.%), a mesoporous catalyst, was higher compared to when using an H-ZSM-5 catalyst (2.2-14-1 wt.%) [40]. Twaiq et al. [41] also found that the amount of coke formed from catalytic pyrolysis of palm oil was higher over MCM-41 compared to ZSM-5 and USY zeolites. With MCM-41, the coke yield was 5-12 wt.%, whereas, with HZSM-5, it was about 2, and 5 wt.% with USY zeolite at same reaction conditions [41]. In this study, the high coke deposition in Al-MCM-41 catalyst might be due to their low acidity and high pore volume, resulting in the inefficient dehydrogenation of pyrolysis vapor and the formation of coke [42]. Aguado et al. [43] added that due to Al-MCM-41 uniform mesoporosity, the formation of coke is feasible in Al-MCM-41 unhindered porous structure [43]. Kim et al. [44] also showed a similar result that the catalytic activity of Al-MCM-41 decreased faster than H-ZSM-5 due to its low acidity and coking. Among the experiments with catalyst mixture, the coke yield was reduced with an increased H-ZSM-5 proportion. The lowest yield among all the catalytic pyrolysis was 2.22 wt.% when using H87.5/A12.5, which suggested that there was a synergistic effect between H-ZSM-5 and Al-MCM-41 catalysts.

  1. In line 200-205, the authors state that coke yield using mesoporous catalyst was higher than that using H-ZSM-5 catalyst, since the H-ZSM-5 constrains coke formation inside the zeolite pores. However, in line 54-57, the authors cited previous literatures demonstrating that mesoporous structures with a large pore size reduce the chances of coke deposition, which contradicts the line 200-205. This needs to be clarified.

Response:

As suggested, the paragraphs have been elaborated (line 204-231).

  1. Line 79-80, “Nevertheless, few recent studies on the mixture of H-ZSM-5 and Al-MCM catalysts and its application on lignocellulose biomass” is not a completed sentence and missing citations.

Response:

The paragraph has been revised and more citations have been added (line 67-86). The authors would like to highlight a knowledge gap in the field of study.

Considering the advantages of H-ZSM-5 and Al-MCM-41, researchers have altered the shape selectivity and porosity of catalysts to allow multi-step/cascade reactions to take place [21–24] in order to render a proper design of catalytic pyrolysis process of biomass and to provide high activity, selectivity, and longer life catalyst. Araujo et al. [25]investigated the catalytic effect of mechanical mixtures of H-ZSM-5 and Al-MCM-41 on the quality of bio-oil from sunflower oil. They found that the mixture of 50% H-ZSM-5 and 50% Al-MCM-41 resulted in a balance carbon fraction between gasoline (C5-C10), kerosene (C11-C15), and diesel (C16-C24) range, corresponding to 44.7%, 42.2%, and 10.5%, respectively [25]. More recently, Li et al. [9] studied the online upgrading of vacuum pyrolysis of rape straw in a two-stage reactor using H-ZSM-5 and MCM-41. They found that 1:1 mixed ratio of H-ZSM-5:MCM-41 was the best among all ratios in their experiments. The oxygen content, H/C, O/C, the high heating value of the bio-oil were 12.81%, 1.701, 0.126, and 34.31 MJ/kg, respectively [9]. Other studies have been carried out on the mixture of microporous and mesoporous catalysts apart from H-ZSM-5 and Al-MCM-41 [25–27]. Nevertheless, the synthesized catalysts have been of limited use on the bench-scale reactor. In industrial production, factors which influence the preference for catalysts in the industry are the ease of separation of products from catalysts [28–30], reusability [29], relatively inexpensive [29,30], and ease to handle [29]. In this study, a simple method is proposed to improve the quality of bio-oil during the biomass catalytic pyrolysis process by physically mixing commercial mesoporous catalyst in pellet form, Al-MCM-41, and microporous catalyst in sphere form, H-ZSM-5. This approach can provide an affordable process that can be easily adapted in industries.

  1. Line 81-86 regarding the composition of Lignocel HBS sample use in this work seems irrelevant in this paragraph. The authors may consider moving these sentences to the materials/biomass section (Line 109-113).

Response:

The sentences regarding the composition of lignocel HBS sample has been removed since it is incoherent, as mentioned by the reviewer. The authors decided not to move it to the material/biomass section either since the authors did not analyse the biomass composition themselves. Further, the authors did not elaborate the effect of catalyst mixture on individual composition of biomass (cellulose, hemicellulose, and lignin). 

  1. Line 46-62, the terms of ZSM-5 and MCM-41, instead of H-ZSM-5 and Al-MCM-41, have been used in these lines.

Response:

The term ZSM-5 has been changed to H-ZSM-5 and the term MCM-41 has been changed to Al-MCM-41 (line 50-66).

Reviewer 2 Report

Review of catalysts-874400

Line 79-80 A quick search yielded quite a few publications on the use of H-ZSM-5 and Al-MCM-41 catalyst mixtures for application in catalytic fast pyrolysis.

Lines 81-86 Why do the authors compare Lignocel composition to that of sunflower oil and rape straw at this point in the manuscript?

Line 109 Why was Lignocel used as a feedstock? More information on the choice of feedstock and why it is important to study should be included in the introduction.

Lines 119 -121 Experiments where only H-ZSM-5 or only Al-MCM-41 were used should be included here.

Lines 140-141 How were the OF and AF segregated? This is important as the method for separation will affect what compounds will be found in each. For example, was centrifugation, gravity separation or simply decantation used?

Line 179 “…the highest OF value was achieved with H87.5/A12.5 catalyst.” The highest OF was actually achieved by NC. The highest OF value among the catalyst mixtures wash achieved with H87.5/A12.5. Please make this distinction.

Table 1 What does NC stand for? Can this be included in footnote beneath Table 1?

Table 2 Where are the standard deviation values for this table?

Line 263 This value should be 1.27 not 1.29. Please check the calculation.

Line 290-291 In the OF the selectivity was greater for toluene, xylene and naphthalene when using H87.5/A12.5 and H75/A25 than when using a single H-ZSM-5 catalyst. The selectivity decreased for benzene. Please review the data in Table 3.

Line 292-293 The statement and the data in Table 3 do not match. Please review this.

Lines 308-314 Make a statement regarding how compounds were determined to be desirable or undesirable and reference any pertinent literature.

Line 321-323 Is what is being observed with respect to the results for mixtures as compared to the single catalysts the result of an additive or synergistic effect?

Line 359-360 & Line 362-363 Again, are these observations with respect to the results for mixtures as compared to the single catalysts the result of an additive or synergistic effect?

In Table 1 the coke yield for H87.5/A12.5 is an example of a synergistic effect between H-ZSM-5 and Al-MCM-41. The coke yield is lower for H87.5/A12.5 than what can be achieved for either catalyst alone. This should be mentioned in the manuscript as well as any other examples of synergistic effects that can be elucidated.

Author Response

  1. Line 79-80 A quick search yielded quite a few publications on the use of H-ZSM-5 and Al-MCM-41 catalyst mixtures for application in catalytic fast pyrolysis.

Response:

The paragraph has been revised and more citations have been added (line 67-86). The authors would like to highlight a knowledge gap in the field of study.

Considering the advantages of H-ZSM-5 and Al-MCM-41, researchers have altered the shape selectivity and porosity of catalysts to allow multi-step/cascade reactions to take place [21–24] in order to render a proper design of catalytic pyrolysis process of biomass and to provide high activity, selectivity, and longer life catalyst. Araujo et al. [25] investigated the catalytic effect of mechanical mixtures of H-ZSM-5 and Al-MCM-41 on the quality of bio-oil from sunflower oil. They found that the mixture of 50% H-ZSM-5 and 50% Al-MCM-41 resulted in a balance carbon fraction between gasoline (C5-C10), kerosene (C11-C15), and diesel (C16-C24) range, corresponding to 44.7%, 42.2%, and 10.5%, respectively [25]. More recently, Li et al. [9] studied the online upgrading of vacuum pyrolysis of rape straw in a two-stage reactor using H-ZSM-5 and MCM-41. They found that 1:1 mixed ratio of H-ZSM-5:MCM-41 was the best among all ratios in their experiments. The oxygen content, H/C, O/C, the high heating value of the bio-oil were 12.81%, 1.701, 0.126, and 34.31 MJ/kg, respectively [9]. Other studies have been carried out on the mixture of microporous and mesoporous catalysts apart from H-ZSM-5 and Al-MCM-41 [25–27]. Nevertheless, the synthesized catalysts have been of limited use on the bench-scale reactor. In industrial production, factors which influence the preference for catalysts in the industry are the ease of separation of products from catalysts [28–30], reusability [29], relatively inexpensive [29,30], and ease to handle [29]. In this study, a simple method is proposed to improve the quality of bio-oil during the biomass catalytic pyrolysis process by physically mixing commercial mesoporous catalyst in pellet form, Al-MCM-41, and microporous catalyst in sphere form, H-ZSM-5. This approach can provide an affordable process that can be easily adapted in industries.

  1. Lines 81-86 Why do the authors compare Lignocel composition to that of sunflower oil and rape straw at this point in the manuscript?

Response:

The sentences regarding the composition of lignocel HBS sample has been removed since it is incoherent, as mentioned by the reviewer. The authors would like to highlight a knowledge gap in the field of study and to the best of author’s knowledge, it has been expressed in line 67-86. 

  1. Line 109 Why was Lignocel used as a feedstock? More information on the choice of feedstock and why it is important to study should be included in the introduction.

Response:

As suggested, more information on the use of lignocel in the study has been added (line 24-28).

As a source of carbon for liquid transportation fuels, biomass must be converted to liquids at high yields and high qualities [1,2]. One of the promising biomass materials for transportation fuel replacement is lignocellulose biomass. The availability of lignocellulose biomass is abundant [3]. It is also known to be the cheapest and fastest-growing of the biomass materials [4]. Further, it is a scalable, economically viable, and potential carbon neutral feedstock for the production of bio-oil [5].  

  1. Lines 119 -121 Experiments where only H-ZSM-5 or only Al-MCM-41 were used should be included here.

Response:

Experiments using H-ZSM-5 and Al-MCM-41 were mentioned, as suggested (line 120-122).

Experiments with single H-ZSM-5 catalyst and single Al-MCM-41 catalyst were also carried out to compare with the experiments with catalyst mixtures.

  1. Lines 140-141 How were the OF and AF segregated? This is important as the method for separation will affect what compounds will be found in each. For example, was centrifugation, gravity separation or simply decantation used?

Response:

More information on separating OF and AF has been added, as suggested (line 141-142).

The two phases within the liquid products, the organic and aqueous phases, were separated using a decanting funnel.

  1. Line 179 “…the highest OF value was achieved with H87.5/A12.5 catalyst.” The highest OF was actually achieved by NC. The highest OF value among the catalyst mixtures wash achieved with H87.5/A12.5. Please make this distinction.

Response:

As suggested, it has stated the highest OF value among the catalytic pyrolysis test with different proportions of H-ZSM-5 and Al-MCM-41 (line 182-184).

Among the catalytic pyrolysis tests with different proportions of H-ZSM-5 and Al-MCM-41, the highest OF value was achieved with H87.5/A12.5 catalyst.

  1. Table 1 What does NC stand for? Can this be included in footnote beneath Table 1?

Response:

‘NC: non-catalytic case’ has been included in the footnote beneath Table 1, as suggested.

  1. Table 2 Where are the standard deviation values for this table?

Response:

The elemental analysis of the OF and AF samples was carried out once outside the laboratory (external institution). Another minor yet substantial reason is due to limited research funding.

  1. Line 263 This value should be 1.27 not 1.29. Please check the calculation.

Response:

It has been re-calculated, and it is 1.27, not 1.29, as mentioned by reviewer.

  1. Line 290-291 In the OF the selectivity was greater for toluene, xylene and naphthalene when using H87.5/A12.5 and H75/A25 than when using a single H-ZSM-5 catalyst. The selectivity decreased for benzene. Please review the data in Table 3.

Response:

The paragraph has been reviewed, as suggested (line 324-330).

The present findings also highlight the selectivity of toluene and xylene, which was higher when using H87.5/A12.5 than when using a single H-ZSM-5 catalyst. Moreover, the highest selectivity of toluene (12.64%) and xylene (26.42%) was obtained when using H87.5/A12.5 catalyst mixture. A synergistic effect was observed for the formation of toluene and xylene, which could be explained by the enhanced Diels-Alder type reactions between furans and olefins from biomass [26,44]. This could be attributed to the cracking effect of Al-MCM-41 on the biomass for H-ZSM-5 conversion. The catalyst mixture of H-ZSM-5 and Al-MCM-41 can alter the selectivity of aromatics in the OF.

  1. Line 292-293 The statement and the data in Table 3 do not match. Please review this.

Response:

As suggested, the paragraph and the numbers mentioned have been revised (line 324-330).

  1. Lines 308-314 Make a statement regarding how compounds were determined to be desirable or undesirable and reference any pertinent literature.

Response:

References on how compounds were grouped as desirable and undesirable compounds have been added, as suggested (line 340-346).

The chemical compounds were grouped as desirable and undesirable compounds to compare how the chemical compounds were distributed between OF and AF. Hydrocarbons, phenols, furans, and alcohols were included as desirable compounds [13,69–73], while other compounds, like acids, carbonyls, carboxylic, ethers, Polycyclic Aromatic hydrocarbons (PAHs), oxygenates, and other unidentified compounds were known to be undesirable compounds [13,69–71,73]. Figure 4shows the proportion of desirable and undesirable compounds in OF for non-catalytic and catalytic pyrolysis as the percentage of total ion chromatogram area.

  1. Line 321-323 what is being observed with respect to the results for mixtures as compared to the single catalysts the result of an additive or synergistic effect?

Response:

The authors showed that the desirable compounds (hydrocarbons, phenols, furans, and alcohols – purple bars in Figure 4) in the oil from experiments using H87.5/A12.5 and H75/A25 was higher than only using single H-ZSM-5 or Al-MCM-41 catalyst. With catalyst mixture H87.5/A12.5 and H75/A25, the desirable compounds were 95.89% and 94.39%, respectively, while with single H-ZSM-5 and Al-MCM-41, they were 93.91% and 81.00%, respectively.

  1. Line 359-360 & Line 362-363 Again, are these observations with respect to the results for mixtures as compared to the single catalysts the result of an additive or synergistic effect?

Response:

The authors provide the BET surface areas for the fresh and spent catalysts to show the effect of coke formation on the catalyst surface. The BET surface area reduction for catalyst mixtures could be identified and compared with the surface area reduction for single catalysts.

  1. In Table 1 the coke yield for H87.5/A12.5 is an example of a synergistic effect between H-ZSM-5 and Al-MCM-41. The coke yield is lower for H87.5/A12.5 than what can be achieved for either catalyst alone. This should be mentioned in the manuscript as well as any other examples of synergistic effects that can be elucidated.

Response:

As suggested, more explanation on coke yield has been added to show the synergistic effect of catalyst mixture (line 204-231).

The coke yield decreased from 3.72 wt.% when using single Al-MCM-41 catalyst to 2.22 wt.% when 87.5 wt.% of H-ZSM-5 catalyst was added in the catalyst mixtures. Then, the coke yield increased to 2.78 wt.% when using single H-ZSM-5 catalyst. The fluctuation of coke yield when using single H-ZSM-5 catalyst was caused by the microporosity structure of H-ZSM-5, which hindered the large molecule oxygenates to enter the pores and caused the growth of coke [9,12,17,18]. In the absence of Al-MCM-41 catalyst, there was no pathway to convert large oxygenates to small ones before reacting with H-ZSM-5 catalyst. However, the yield of coke in Al-MCM-41 catalyst was 0.94 wt.% higher than it in H-ZSM-5 catalyst. This result is contradicted by the experiments of Zhang et al. [26] on catalytic pyrolysis of rice stalk using LOSA-1, which mainly composed of ZSM-5, and Gamma-Al2O3. The coke yield in their experiment decreased from 30.3% with pure LOSA-1 to 23.5% with 50% Gamma-Al2O3. Gamma-Al2O3, as a mesoporous catalyst, had strong cracking characteristic and can convert the large-molecule oxygenates into small-molecule oxygenates. Thus, the oxygenates molecules can leave the catalyst after their formation before they polymerize and lead to coke [39].    

Nonetheless, the current finding is consistent with the results of past studies by Adjaye et al. [40]. They observed that the coke resulting from a biofuel conversion, when using a silica-alumina (6.8 – 27.9 wt.%), a mesoporous catalyst, was higher compared to when using an H-ZSM-5 catalyst (2.2-14-1 wt.%) [40]. Twaiq et al. [41] also found that the amount of coke formed from catalytic pyrolysis of palm oil was higher over MCM-41 compared to ZSM-5 and USY zeolites. With MCM-41, the coke yield was 5-12 wt.%, whereas, with HZSM-5, it was about 2, and 5 wt.% with USY zeolite at same reaction conditions [41]. In this study, the high coke deposition in Al-MCM-41 catalyst might be due to their low acidity and high pore volume, resulting in the inefficient dehydrogenation of pyrolysis vapor and the formation of coke [42]. Aguado et al. [43] added that due to Al-MCM-41 uniform mesoporosity, the formation of coke is feasible in Al-MCM-41 unhindered porous structure [43]. Kim et al. [44] also showed a similar result that the catalytic activity of Al-MCM-41 decreased faster than H-ZSM-5 due to its low acidity and coking. Among the experiments with catalyst mixture, the coke yield was reduced with an increased H-ZSM-5 proportion. The lowest yield among all the catalytic pyrolysis was 2.22 wt.% when using H87.5/A12.5, which suggested that there was a synergistic effect between H-ZSM-5 and Al-MCM-41 catalysts.

Further, other synergistic effects have been explained in the article. Below is the summary of bio-oil quality from catalytic pyrolysis using H87.5/A12.5 as compared to when using single H-ZSM-5 catalyst. The amount of organic fraction from catalytic pyrolysis using H87.5/A12.5 is the second best. However, in terms of oxygen content, O/C molar ratio, H/C molar ratio, HHV, deoxygenation degree, and water content, the quality of bio-oil from catalytic pyrolysis using H87.5/A12.5 catalyst is better than the bio-oil from the process using single H-ZSM-5 catalyst. The quality of bio-oil from the process using other catalyst mixtures and single Al-MCM-41 is also exceptional. Yet, the yield of organic fraction is low, which is less preferable since this study aims to produce bio-oil as a liquid fuel or a drop-in feedstock in existing refinery.       

Parameter

H87.5/A12.5 Bio-Oil

H-ZSM-5 Bio-Oil

Oxygen content (wt.%)

15

21

O/C molar ratio

0.21

0.15

H/C molar ratio

1.26

1.27

HHV (MJ/kg)

34.15

32.28

Deoxygenation degree (%)

65.75

52.50

Water content (wt.%)

0.39

3.44

Organic fraction (wt.%)

5.66

7.08

Desirable compounds (%)

95.89

93.91

Reviewer 3 Report

Peer Review Report

Effect of H-ZSM-5 and Al-MCM-41 Proportions in Catalyst Mixtures on the Composition of Bio-Oil in Ex-Situ Catalytic Pyrolysis of Lignocellulose Biomass

Comments to Author:

Overview and general recommendation:

The study at hand is of interest to the scientific community researching lignocellulose biomass. Converting bio-oil into a more usable form through de-oxygenation is attractive across many applications. Optimization of the catalysts used in the pyrolysis of this becomes paramount which this paper highlights.

The paper overall contains a good analysis of the methodologies and findings which sound recommendations. Therefore, I recommend that it be accepted with minor comments/revisions.  

Major comments

There are no major comments for this article.

Minor comments/revisions

  1. In the last sentence of the abstract, “utilisation” should be “utilization.”
  2. Line 189: how were the error bars (uncertainties) determined in Table 1?
  3. Line 273-274: Can the authors expand on why an increase in the oxygen content leads to a decrease of the HHV?
  4. Line 275: insert “in” between ”interferes” and “the”.
  5. Line 290: Can the authors supply a reason as to why BTX production is higher in H87.5/A12.5 and H75/A25 compared to a single catalyst?  

Author Response

The study at hand is of interest to the scientific community researching lignocellulose biomass. Converting bio-oil into a more usable form through de-oxygenation is attractive across many applications. Optimization of the catalysts used in the pyrolysis of this becomes paramount which this paper highlights.

The paper overall contains a good analysis of the methodologies and findings which sound recommendations. Therefore, I recommend that it be accepted with minor comments/revisions. 

  1. In the last sentence of the abstract, “utilisation” should be “utilization.”

Response:

The word has been corrected, as suggested.

  1. Line 189: how were the error bars (uncertainties) determined in Table 1?

Response:

More information on how the error bars were determined in Table 1 has been added to the paragraph (line 178-181).

The error bars represent standard deviations in absolute % for all experiments. The standard deviations were calculated by Excel with non-biased or n-1 method [35]. The scatter in the product yields is always less than 5%, indicating that the reproducibility is sufficient for observing trends in all experiments.  

  1. Line 273-274: Can the authors expand on why an increase in the oxygen content leads to a decrease of the HHV?

Response:

More information and references have been added to expand the explanation and support the study (line 299-301).

It is generally considered that the increase of the oxygen content leads to a decrease of the HHV [57–60]. The oxygen content in fuel will reduce the ability of the fuel to burn, resulting in less amount of heat produced and low heating value.  

  1. Line 275: insert “in” between ”interferes” and “the”.

Response:

The sentence has been corrected, as suggested.

  1. Line 290: Can the authors supply a reason as to why BTX production is higher in H87.5/A12.5 and H75/A25 compared to a single catalyst?

Response:

More information and references have been added, as suggested (line 317-330).

The chemical compositions of liquids from non-catalytic and catalytic pyrolysis of lignocellulose biomass are summarised in Table 3. In the analysis of OF, the lowest selectivity of benzene (10.91%), toluene (7.20%), and xylene (7.48%) were obtained when using a single Al-MCM-41 catalyst. The BTX yields were correlated with strong acid contents of the catalysts [32,64], resulting in the high catalytic activity in aromatization [65]. The data in Table 3 indicates that H-ZSM-5 had higher strong acid content than Al-MCM-41. Therefore, among the experiments with catalyst mixture, the selectivity of BTX decreased with a further decreased proportion of H-ZSM-5 in the catalyst mixture.    

The present findings also highlight the selectivity of toluene and xylene, which was higher when using H87.5/A12.5 than when using a single H-ZSM-5 catalyst. Moreover, the highest selectivity of toluene (12.64%) and xylene (26.42%) was obtained when using H87.5/A12.5 catalyst mixture. A synergistic effect was observed for the formation of toluene and xylene, which could be explained by the enhanced Diels-Alder type reactions between furans and olefins from biomass [26,44]. This could be attributed to the cracking effect of Al-MCM-41 on the biomass for H-ZSM-5 conversion. The catalyst mixture of H-ZSM-5 and Al-MCM-41 can alter the selectivity of aromatics in the OF.
